# The Cat–Owner Relationship: Validation of the Italian C/DORS for Cat Owners and Correlation with the LAPS

**DOI:** 10.3390/ani13010069

**Published:** 2022-12-24

**Authors:** Carmen Borrelli, Giacomo Riggio, Tiffani Josey Howell, Patrizia Piotti, Silvana Diverio, Mariangela Albertini, Paolo Mongillo, Lieta Marinelli, Paolo Baragli, Francesco Paolo Di Iacovo, Angelo Gazzano, Federica Pirrone, Chiara Mariti

**Affiliations:** 1Department of Veterinary Sciences, University of Pisa, 56124 Pisa, Italy; 2Anthrozoology Research Group, School of Psychology and Public Health, La Trobe University, Bendigo, VIC 3552, Australia; 3Department of Veterinary Medicine and Animal Sciences, University of Milan, 26900 Lodi, Italy; 4Laboratory of Ethology and Animal Welfare (LEBA), Department of Veterinary Medicine, University of Perugia, 06126 Perugia, Italy; 5Laboratory of Applied Ethology, Department of Comparative Biomedicine and Food Science, Università degli Studi di Padova, Viale dell′Università 16, 35020 Legnaro, Italy

**Keywords:** Lexington attachment to pets scale, MDORS, Monash dog owner relationship scale, C/DORS, cat–owner bond

## Abstract

**Simple Summary:**

Since most people globally own a pet, it is important to understand and assess the features of the owner–pet relationship. The Cat-/Dog-Owner Relationship Scale (C/DORS) is a questionnaire aimed at investigating specific aspect of pet-owner relationships. This tool has never been tested for validity or reliability on cat owners in Italy. We achieved this aim by translating and back translating the C/DORS from the original English into Italian and modifying the original response options from 1–5 to 1–7 to increase variability in responses. We determined the most appropriate factor model, which was the same one reported for the English version (i.e., Perceived Emotional Closeness = PEC, Pet–Owner Interactions = POI, Perceived Costs = PC). We confirmed its construct validity by correlating the C/DORS subscales with the Lexington Attachment to Pets (LAPS) subscales. We found a higher score for POI and PEC for those cats living exclusively indoors compared with indoor/outdoor cats. People that also owned a dog scored lower in POI but higher in PC. Similarly, behaviour problems, not being neutered, and lack of previous experience with cat ownership were associated with lower scores on PC.

**Abstract:**

Globally, most people now own a pet. Scales have been developed to understand the impact of pet ownership on people’s lives and to measure specific aspects of the owner–pet relationship. The Cat-/Dog-Owner Relationship Scale (C/DORS) is a tool developed to investigate this relationship in both dog and cat owners. The aim of the study was to refine and validate the C/DORS for cat owners in Italian. Exploratory Factor Analysis and Confirmatory Factor Analysis (CFA) were used to determine the most appropriate factor model. Construct validity was confirmed by correlating the C/DORS subscales with the Lexington Attachment to Pets (LAPS) subscales. Results confirmed the original structure of the English version (i.e., three factors: Pet Owner Interaction = POI, Perceived Emotional Closeness = PEC, Perceived Costs = PC) and CFA confirmed the structure of LAPS and C/DORS scales. Cronbach’s alpha demonstrated the Italian version of the two scales to have good internal reliability for all domains. Owners of cats living exclusively indoors reported higher scores on POI and PEC compared to indoor/outdoor cats. Owning both cats and dogs was correlated with lower scores on POI, and fewer perceived costs (i.e., PC) of cat ownership. Finally, behaviour problems, not being neutered, and lack of previous experience with cat ownership were associated with lower scores on PC.

## 1. Introduction

Estimates from 2016 indicate that a majority of people globally own a pet [1], so it is important to understand the impacts of pet ownership on people’s lives. The psychological and physical health benefits of pet ownership have been studied since the 1980s [2]. However, the results are often conflicting, with some studies finding benefits for pet owners, and others finding no effect or even a negative effect [3,4]. How ‘ownership’ is defined could be a factor explaining these mixed results, because a comparison of pet owners versus non-owners may not include the frequency and nature of interactions between the owner and pet, the strength or style of pet-owner attachment, or even how long the pair have lived together [5]. Therefore, scales have been developed to attempt to measure specific aspects of the pet–owner relationship. Some commonly used scales include the Comfort from Companion Animals Scale [6], the Companion Animal Bonding Scale [7], the Pet Attachment Questionnaire [8], and the Lexington Attachment to Pets Scale (LAPS) [9].

### 1.1. Lexington Attachment to Pets Scale

One of the most popular scales used to measure the quality of the pet–owner attachment is the LAPS [9], and it has been used with many species. Indeed, in one study alone, the authors included owners of dogs, cats, birds, horses, fish, reptiles, rabbits, rodents, and livestock animals [10]. The LAPS has been validated in the original English [9] and translated into other languages (e.g., German [11], Spanish [12], Italian [13]) but the scale focuses exclusively on the affective aspects of the pet–owner relationship. This was intentional [9], as it helped reduce the dog bias inherent in previous scales which asked questions about certain types of pet–owner interactions (e.g., ‘I take my pet along when I go jogging or walking’ on the Pet Relationship Scale [14]). However, it required limiting the scale to one aspect of the pet–owner relationship, excluding any consideration of the interactions that people have with their pets, or the negative elements (e.g., the financial and time costs) of pet ownership. Furthermore, in being designed for use with any pet type, the LAPS is unable to describe variations characterising pet–owner relationships among different species. For example, a dog–owner relationship is likely to be different from a cat–owner relationship. Dog owners may be more likely to take their dog on outings away from their home than cat owners. On the other hand, cat owners have indicated that they like their cat’s independence [15], which may be less desirable in a dog. Unfortunately, the LAPS is unable to detect these subtle differences, and species-specific scales would be useful.

### 1.2. Monash Dog–Owner Relationship Scale

The Monash Dog–Owner Relationship Scale (MDORS) was developed in an attempt to address some of the limitations of existing scales, such as lack of species specificity or a focus on just one aspect of the pet–owner relationship [16]. The MDORS is designed for use in dog owners, and it examines both the positive and negative aspects of dog ownership [16]. It is based on Social Exchange Theory, which posits that a relationship can be considered good when the positive aspects outweigh the negative ones, rather than ignoring any negative aspects altogether [17]. The MDORS includes three subscales: Dog-Owner Interactions (DOI), Perceived Emotional Closeness (PEC), and Perceived Costs (PC). DOI captures the frequency of shared activities that owners have with their dogs. PEC includes items related to the affective elements of dog–owner relationships, such as the items in the LAPS. PC relates to financial, time, and emotional costs of dog ownership (i.e., the more negative aspects of having a dog) [16]. The MDORS represents a theoretical advancement in this area because it acknowledges that there are costs associated with dog ownership but accepts that the presence of negative aspects of a relationship does not necessarily equate to a poor-quality relationship overall.

While the MDORS is a useful addition to the available pet–owner relationship scales, it has proved difficult to translate in some languages, limiting its generalisability outside of the English-speaking world. For instance, a German translation found that a five-subscale model was more appropriate for their sample [18], rather than the three subscales used in the original English version. Furthermore, a Spanish translation found a two-factor model, effectively combining DOI and PEC into one subscale [19]. A Danish translation performed a reliability analysis on the three existing subscales, and found acceptable Cronbach’s alpha levels for two of the three subscales, with an alpha score of only 0.60 for DOI [20]. The full, original MDORS has not been translated into Italian, but individual items have been translated and used in previous research [21].

### 1.3. Cat–Owner Relationship Scale

Having a scale specific to dog-owner relationships is useful because dogs are a commonly owned pet type. However, cats are also one of the most popular pet animals in the world [1], so the Cat Owner Relationship Scale (CORS) was developed to measure the same aspects of the cat–owner relationship that the MDORS measures for dog-owner relationships [15]. The CORS was based on the MDORS, but some items were added based on consultations with cat owners, who described key components of cat–owner relationships that are less relevant to dog-owner relationships (e.g., appreciating the animal’s independence). At the end of the CORS, the authors provided a Cat- and Dog-Owner Relationship Scale (C/DORS), which includes the full CORS but with different scoring instructions for cats and dogs. The C/DORS has never been validated in dogs in English. 

Recently, Riggio et al. [22] refined and validated the C/DORS for dogs in Italian. The refinement is very similar to the original C/DORS, but with response options on a scale of 1–7, rather than the original 1–5, to increase variability. It also includes additional items that were included in the CORS [15] but not in the original MDORS. However, the refined C/DORS has not yet been validated in cats. It would be useful to validate this scale in both dogs and cats, because it may then be theoretically possible to compare cat and dog owners on a scale developed for use in both species, despite the inherent differences in cat- and dog-owner relationships. 

### 1.4. The Current Study

The aim of this study was to refine and validate the C/DORS for cat owners in Italian. We achieved this aim by translating and back translating the C/DORS from the original English into Italian and then modifying the original response options from 1–5 to 1–7 to increase variability (Depending on the question 1 could be “I completely disagree” or “once per month or less” while 7 could be “I completely agree” or “several times per hour”). We then used Exploratory Factor Analysis and Confirmatory Factor Analysis to determine the most appropriate factor model. We confirmed its construct validity by correlating the C/DORS subscales with the LAPS subscales, and we confirmed test–retest reliability with a subsample of participants. 

## 2. Materials and Methods

The study received the approval from the Ethical Committee of the University of Pisa, Italy (protocol n 5/2022 in accordance with Directive 2010/63/EU).

### 2.1. Questionnaire Description

The questionnaire comprised 5 different sections for a total of 78 items. The first two sections asked about demographic data regarding the owner (i.e., age, gender, employment status, living environment, previous cat ownership experience) and the cat (i.e., current age, age when adopted, sex, breed, neuter status, physical, and behavioural problems). The third section consisted of an updated version of the C/DORS [15] already used for dog owners in a previous study by Riggio et al. [22]. The fourth section contained the 23 items of the LAPS [9]. Finally, we asked the owners to describe both their cat and their relationship with the cat, using three adjectives. Additionally, the owners were asked to rate their perception regarding the strength of their cat’s bond towards them. The questionnaire was administered in Italian. The full questionnaire in the Italian and English version is available in the Appendix B. (Table A1: Original questionnaire in Italian and in English).

### 2.2. Questionnaire Refinement 

This questionnaire was already used in a previous study by Riggio et al. [22], but since we wanted to address only cat owners in the current study, the word “dog” was replaced with “cat” in all sections. Additionally, the question “What kinds of activities does your pet do?” was replaced with “Does your cat come with you when you go for a walk?”. 

### 2.3. Questionnaire Distribution 

Using Jotform^®^, an online version of the questionnaire was developed and distributed through an electronic link on Facebook^®^, Menlo Park, CA, USA (www.facebook.com accessed on 20 March 2022). Responses were collected between March and April 2022. To complete the questionnaire, respondents had to (1) be cat owners, (2) be at least 18 years old and (3) have agreed to the informed consent. If they owned more than one cat, they were asked to consider the one they had been living with for the longest time.

### 2.4. Test–Retest Reliability

To assess test–retest reliability, 30 cat owners were asked to complete the questionnaire a second time, about 30 days after the first completion. To be able to match questionnaires completed by the same respondents without gathering information about their identity, researchers asked participants to write down a personal code (both during the first and second time of participation) and report whether it was the first or the second time they were filling the questionnaire out, as well as the number of days that passed.

### 2.5. Statistical Analyses 

The statistical analyses were performed using R statistical software [23]. The packages psych [24], GPArotation [25], and lavaan [26] were used for factor analyses, and the packages ordinal [27], rcompanion [28], and emmeans [29] for regressions. 

For some variables, response options were aggregated for statistical analysis. Specifically, the cat’s breed was maintained if it was a breed reported at least five times; all other breeds were renamed as “other” to reduce the complexity of the model. For the same reason, the presence of other animals in the house was summarized in four categories: no other pets, dogs only, other cats only, both cats and dogs. The source of the pet was divided based on four categories: those places where the pet can develop behavioural problems or can have poor socialization, such as pet shops [30]; those places where the previous experiences are usually unknown as shelters/pets found in the streets, and those acquired by private individuals (e.g., friends, non-professional breeders), regulated breeders or born at home; in addition, cats given as a present were classified as a separate category.

Variables in which respondents could choose more than one and/or could add free text were also categorised into groups. For example, the owners’ employment status was organized based on whether they worked with animals or not; other categories were freelancer, pensioner, and student. For the question “Does your cat come with you when you go for a walk?” the responses were condensed into five categories: no, yes on a leash, yes off leash, yes while in the garden, other containment methods. Finally, the two questions regarding the presence of health conditions and behavioural problems were categorised into binary data (i.e., yes/no).

Data from the C/DORS were assessed in preparation for the explanatory factor analysis (EFA) used to examine the factor structure of the scale. The items were assessed to exclude those with low standard deviation (SD < 0.5), high skewness and kurtosis (values > 6). The items: “How often do you feel looking after your cat is a chore?” (item n.14 kurtosis = 11.36); “How often does your cat stops you from doing things you want to?” (item n.16 kurtosis = 21.73); “How often do you feel that having a cat is more trouble than it’s worth?” (item n19 skewness = 12.03, kurtosis = 173.03; “How often do you take your cat to visit people?” (item n.27 kurtosis = 25.90); “How often do you take your cat in the car?” (item n.29 kurtosis = 9.96) were excluded from further analysis. Thus, 27 items were entered into the EFA. The results of the parallel analysis and the scree plot suggested a three-factor solution. Therefore, three factors were extracted with an oblimin rotation and factor loading cut-off of 0.3. The oblique rotation was chosen because we hypothesised a degree of overlapping between factors. Three items (i.e., “How often do you buy presents for your cat?”, “How often do you give treats to your cat?”, “How often do you groom your cat?”) were excluded from the analysis because their loading was below the threshold. The results with loadings and descriptive statistics of the removed items are in the Appendix A. Therefore, the remaining 24 items were analysed with another EFA with oblimin rotation.

Subsequently, we investigated the item response distributions of the LAPS and C/DORS questionnaires performing a confirmatory factor analysis (CFA). Given the ordinal nature of the data, we used the diagonally weighted least squares (DWLS) robust estimator. Several fit indices were considered to evaluate models: comparative fit index (CFI), Tucker–Lewis Index (TLI), root-mean-square error of approximation (RMSEA). Cut-off values for adequate fit for CFI and TLI were >0.90, RMSEA < 0.06 [31]. The descriptive statistics and normality tests of the scales are reported in Table 1. The reliability of both scales was also evaluated measuring the Cronbach’s alpha (estimate of internal reliability). Finally, Spearman rho tests were used to assess test–retest reliability for the LAPS and the C/DORS by looking at correlations between T0 and T1 for each subdomain of the two scales. In addition, we measured whether the C/DORS subscales correlated with the score of the scale overall. To better understand the relationship between the two scales, we investigated the correlations between the LAPS and C/DORS domains.

We used ordinal regressions to investigate the relationship between C/DORS scores and the explanatory variables: owner demographics (i.e., age and gender, level of education, job, household composition, previous experience with dogs) and cat’s characteristics (i.e., signalment, health and behaviour anamnesis, housing situation, and presence of other pets). Before calculating the regression models, we used inferential tests to reduce model complexity. For each domain, a Kruskal–Wallis test was used with categorical data, a Mann–Whitney U test was used with binomial data, and a Spearman rho correlation was used with continuous data in order to identify significant relationships between the given variable and the sub-domain. Alpha level was corrected with Bonferroni corrections for 22 multiple comparisons (α < 0.001). 

Finally, ordinal logistic regression (OLR) models were developed to assess the relationship that Shared activities, Emotional Closeness, and Perceived Costs scores (dependent variables, Y) had with each of the potential predictors (independent variables, X) for which there were significant differences in the preliminary analysis. The factors initially included in the model following inferential analysis were further reduced using a step-one method. Model fitting was tested using a likelihood ratio test and measured using the pseudo R2 calculated with Nagelkerke method, and we calculated pair-wise analyses with a Tukey correction for the categorical predictors to measure the strength of the relationship. A two-sided *p* < 0.05 was considered statistically significant. 

## 3. Results

### 3.1. Demographic Data 

#### 3.1.1. Participants 

One thousand one hundred and eighty-three cat owners participated in the study through the distributed link and completed the questionnaire without missing any responses in the C/DORS and LAPS sections. In addition, 30 cat owners filled the questionnaire twice, and provided a matchable code for test–retest reliability analysis.

The sample was composed mainly of women (88%). Participants were aged 20–75 years old (Mdn = 45). Respondents lived mostly in big cities (58%) with a small sample living in towns (32%), and their level of education was mainly high, with most participants achieving the tertiary degree and above (62%). Most of them did not work with animals (77%). The respondents’ previous experience with cats varied, with most of them having owned at least one cat previously (81%). Since some of the demographic questions were not answered by all participants, the total number of responses for each question reported in Table 2 may vary from the total sample. 

#### 3.1.2. Cat Demographics 

The cat population (more information are available in Appendix A) was equally composed of female (52.2%) and male (47.8%) of 0.5–25 years old (Mdn = 8), with mostly neutered cats (96.4%). Most of the cats were mixed breed (90.5%) with some representations of purebred. Most of them were acquired from a shelter (60%) with a smaller percentage obtained by private individuals (29.3%). The cats had access both inside the house and in the garden (40.1%) or in the house with the possibility of using terraces and roofs (30%) while 29.1% of them were kept exclusively inside the house. Most of them did not report any behavioural (90.1%) or medical issues (78.4%). Cats were not used to walking with their owners (82.5%) except for a small percentage that were accustomed to following the owner outside without a leash (13%). The distribution of cats living alone or with other pets (including other cats or dogs) was quite similar.

### 3.2. Factor Analyses and Questionnaire Validation 

Several widely accepted criteria for the factorability of a correlation were used to assess the factor solution of the C/DORS. Firstly, we checked for outliers looking at standard deviation, skewness, and kurtosis, and for low variance, and we removed items 14, 16, 19, 27 and 29. The Kaiser–Meyer–Olkin measure of sampling adequacy was 0.91, well over the generally recommended value of 0.60, Bartlett’s test of sphericity was significant (χ^2^ (351) = 12,323.54, *p* < 0.001). Given the indicators above, factor analysis was found appropriate for the remaining 26 items. Parallel Analysis (PA) suggested a three -factor solution, explaining 38.3 % of the variance. In the first analysis, almost all items had primary loadings over 0.3, while items with low loadings were removed (items 12, 28 and 31) and then the analysis repeated. The factor loading matrix for this final analysis is reported in Table 3. The factor labels given by Howell et al. [15] were appropriate for the retrieved factors and therefore were used. Factor 1 of the C/DORS included elements that reflected the characteristic of Pet–Owner Interactions (POI), such as spending time together, playing and cuddling. A higher score in this subdimension suggested that the cat and the owner shared more activities. Factor 2 included elements from the Perceived Emotional Closeness (PEC) subdimension, associated with social support, bonding, companionship, and unconditional love. A higher score in this aspect suggested that the owner felt more emotionally connected to their cat. Finally, factor 3 included elements reflecting the facet of Perceived Costs (PC), dealing with negative aspects, such as monetary expenses, responsibilities towards the pet, and possible limitations that the owner will face due to owning a pet. All the elements in this component were reversed, so a higher score indicated that the owner was less influenced by the negative aspects of cat ownership.

CFAs confirmed the structure of LAPS (χ^2^ (df = 227) = 735.32, *p* < 0.001, CFI = 0.99, TLI = 0.98, RMSEA = 0.04) and C/DORS (χ^2^ (df = 227) = 682.63, *p* < 0.001, CFI = 0.98, TLI = 0.97, RMSEA = 0.04) scales. Furthermore, Cronbach’s alpha demonstrated the Italian version of the LAPS (General Attachment α = 0.86; People substitute α = 0.86; Animal Rights α = 0.78) and C/DORS (Pet–Owner Interactions α = 0.84; Perceived Emotional Closeness α = 0.85; Perceived Costs α = 0.71) scales to have good internal reliability for all domains.

Test–retest reliability (n = 30) was found to be strong for the C/DORS PC and C/DORS POI domains, and for the LAPS People Substitute. Test–retest reliability was very strong for the C/DORS PEC domain, and the LAPS General Attachment and LAPS Animal Rights domains (Table 4). Lastly, the overall C/DORS score (Mdn = 104, range = 51–140) had a very strong positive correlation with the POI (rho = 0.80, *p* < 0.001) and PEC (rho= 0.89, *p* < 0.001) domains, and a negative correlation with the PC domain (rho = − 0.05, *p* = 0.06).

### 3.3. Correlation between LAPS and CDORS

We found positive correlations between the LAPS and C/DORS scales (Table 5). Specifically, the C/DORS POI was strongly correlated with the LAPS’ General Attachment and People Substitute, and moderately correlated with Animal Rights. The C/DORS PEC report a strong correlation with all three LAPS subscales. The C/DORS PC showed a weak correlation with all LAPS subscales.

### 3.4. Regressions

The final model calculated for the dependent variable C/DORS POI accounted for 18% of the variance in this domain (Model fit: Nagelkerke Rp2 = 0.18, *p* < 0.001). A main effect without interaction of the fixed factors “Living space” (indoors, outdoors only, indoors and outdoors) “Other pets” (living alone, only with dogs, only with cats, both dogs and cats) and “Perceived cat’s attachment” (very much, much, I don’t know, none, a little, very little) was found (AIC = 6216.61, χ^2^_14_ = 198.29, *p* < 0.001, Table 6). Living both indoors and outdoors decreased the odds of scoring high in the POI dimension, compared to live exclusively inside. Concerning the “Other pets” factors, results of the regression showed that living with both dogs and cats rather than alone or with only other cats and living only with other dogs rather than only with other cats decreased the probability of scoring high in the POI dimensions. Conversely, living alone rather than with only other dogs increased the odds of scoring high in the POI subscale (Table 7).

The model developed for the dependent variable C/DORS PEC accounted for 30.6% of the variance in this domain (Model fit: Nagelkerke Rp2 = 0.31, *p* < 0.001). A main effect without interaction of the fixed factors “Living Space” (indoors, outdoors only, indoors and outdoors), “Household” (1, 2, 3, 4 or more) and “Perceived cat’s attachment” (very much, much, I don’t know, none, a little, very little) was observed (AIC = 6846.24, χ^2^
_14_ = 362.39, *p* < 0.001, Table 6). Living indoors only significantly increased the cat’s probability of scoring high in the PEC dimension, compared to living both inside and outside (e.g., in the house and in the garden) (Table 7).

The model developed for the dependent variable C/DORS PC accounted for 12% of the variance in this domain (Model fit: Nagelkerke Rp2 = 0.12, *p* < 0.001). A main effect without interaction of the fixed factors “Number of previous cats” (0, 1, 2, 3, 4 or more), “Presence of other pets” (living alone, only with dogs, only with cats, both dogs and cats), “Presence of behavioural problems” (yes, no) and “Perceived cat’s attachment” (very much, much, I don’t know, enough, a little, very little) was found (AIC = 5036.73, χ^2^
_17_ = 121.68, *p* < 0.001, Table 6). Owners who had not had a previous cat or who had just one cat were more likely to perceive that taking care of their pet was expensive, compared to owners who had had four or more cats in the past. Similarly, owners perceived higher costs for unneutered cats compared to owners of neutered cats. Conversely, owners that owned a female cat were more likely to score high in the PC dimension, meaning that they perceive maintaining their pet as less expensive, compared to owners who owned a male cat. Concerning the “Presence of other pets” factor, owners of cats that lived alone had lower scores on the PC dimension compared to cats that lived with only other dogs. Similarly, living only with other dogs was associated with higher PC scores compared to cats that lived only with other cats. Finally, owners that reported their cat to have behavioural problems were more likely to score low in the PC dimension, indicating more perceived costs, compared to owners that did not report any behaviour issue (Table 7).

## 4. Discussion

The C/DORS scale was developed from the combination of the MDORS [16] and the CORS [15] scales aimed at investigating the relationship between owners and their pet cat or dog. The advantage of this recent scale is that researchers could use it in multispecies studies allowing the comparison between cats and dogs. The C/DORS scale has been previously translated and validated in Italian with dog owners [22]. Starting from the work by Riggio et al. [22] we used the same version of the questionnaire with the aim of refining and validating it in Italian for cat owners. The Italian C/DORS used in the current study included the same factor reported for the English version of the CORS [15], i.e., Pet–Owner Interactions (POI), Perceived Emotional Closeness (PEC) and Perceived Costs (PC). Nevertheless, several items from the original CORS (i.e., items 12, 14, 16, 19, 27, 28, 29, 31) were removed from the study since they did not fit the final model for the current sample.

One of the aim of the C/DORS scale as it was developed [15] is for researchers to be able to use it for cross-species comparison. In this study, the decision was made to include all the items of the scale (even the dog-specific ones, i.e., items 4, 27, 28, 29, 30, 31 and 32). As already discussed above, some of these items were removed due to their high kurtosis (items 27 and 29) or low loading (items 28 and 31); this might represent that at least for the Italian version, those items are not valid for cats. Concerning the other items (4, 30 and 32), when applied to cat owners, they all loaded on the same factors of dog owners which are POI (items 4 and 30) and PEC (item 32), meaning that they might be valid for both dog and cat owners. Nevertheless, further research is needed to confirm or confute these hypotheses.

Items 14 “How often do you feel that looking after your cat is a chore?”, 16 “How often does your cat stop you doing things you want to?” and 19 “How often do you feel that having a cat is more trouble than it’s worth?” were removed before analyses because of their high kurtosis. In all three cases the majority of the owners choose the option with the lowest frequencies (meaning that on a 1 to 7 scale they selected 1). It is unclear why these items were less relevant to Italian cat owners than English-speaker cat owners. A similar result was also found in Riggio et al. [22], where item 19 was removed due to its high skewness. Even in this case, the authors believed that, in general, owners tend to perceive their relationship with their dogs as more beneficial than detrimental. Nevertheless, they also highlighted that the sample might be biased towards owners that were more willing to complete a questionnaire about their dog and therefore more careful about their mutual relationship. This also might be the case for the current study, having been distributed through online social media.

Items 27 “How often do you take your cat to visit people?” and 29 “How often do you take your cat in the car?” were also removed before analyses due to their high kurtosis. Most of the replies reported the lowest frequency (meaning that on a 1 to 7 scale they selected 1). These items were included in the original MDORS for dog owners, but not the final CORS for cat owners. They are in the C/DORS because this scale represents a combination of both, although the recommended scoring instructions vary by species, with some items included only in the scoring instructions for one species but not the other. Cats might be sensitive to unfamiliar objects/places [32].. Therefore, exposure to a novel environment [32] as well as travelling by car might cause stress in cats [33]. Additionally, cat management is quite different from dog management. First of all, people are not used to taking their cats out on a leash. In fact, 82.5% of owners reported that they do not take their cats on walks with them, and only 3.9% indicated that they walk with their cat on a leash. Consequently, cats are less likely to be taken out to visit other people (such as friends or relatives). Secondly, we should consider that many cats are used to live both inside and outside the house (40% of the cats in the current study) and they can freely decide to go outside on their own. For the same reasons, it seems unlikely for owners to decide to take their cats in the car, unless they are bringing them to the vet, which might happen once or twice per year [33].

Items 12 “How often do you buy your cat presents?” and 28 “How often do you give your cat food treats?”, and Item 31 “How often do you groom your cat?” were removed because they scored below the cut-off point of 0.3 in the factor analysis. These were also not included in the original CORS but they were in the MDORS and are therefore included in the CDORS. Concerning items 12 and 28, it might be that cat owners are less likely to give presents or treats to their pet compared with dog owners. For example, giving treats, in owners’ perception, can be linked with activities like training; since cats are perceived as animals that do not need to be trained, this may be why our sample rarely gave treats to their cats. Similarly, pet gifts are often toys. In the current study, many cats lived both inside and outside the house (40%) so it is possible that they tend to play mostly outside and consequently are less likely to engage in playing activities with their owners once they are home. In the case of item 31, it is likely that owners tend to groom their cats infrequently. First of all, cats are used to grooming themselves to remove ectoparasites [34,35] and dirt, as well as maintain the insulating capacity of the pelage and controlling the body temperature [36]. Secondly, it is possible that most of the subjects of this study were shorthair cats.

We analysed the internal consistency of each subscales, which were very good (Pet–Owner Interactions α = 0.84; Perceived Emotional Closeness α = 0.85; Perceived Costs α = 0.71) and this is consistent with a previous study on dog owners conducted by Riggio et al. [22]. Another aim was to assess test–retest reliability of the C/DORS scale for cat owners; we found it to be strong for all subscales (see Table 4).

To confirm the construct validity of the C/DORS’ subscales we correlated them with the LAPS subscales. The PEC subscale had the highest correlation with all three subscales of the LAPS, and this is consistent with the construct reported by Johnson et al. [9]; that is, the LAPS tends to emphasize the emotional and attachment aspects of pet ownership, like the PEC subscale does. On the contrary, the PC subscale was the one that had the lowest correlations with the LAPS since it reported about the perceived financial and time costs of the relationship, which are not considered in the LAPS.

The regression analysis showed that some demographic factors significantly influenced scores on the C/DORS subscales. For POI and PEC, living exclusively indoors was associated with higher scores on both subscales, compared to cats living both indoors and outdoors. This makes sense conceptually, because spending more time indoors would permit the owner and cat to interact more frequently. Cat owners who keep their cat indoors may also feel compelled to provide additional environmental enrichment for their cat, including playing with the cat for social enrichment, to prevent the cat from becoming bored due to spending all their time inside the house. RSPCA Australia [37] and the American Veterinary Medical Association [38] recommend keeping cats indoors, but also providing plenty of environmental enrichment. Indoor cats with ample opportunities for mental stimulation may experience fewer behavioural problems [39], and owners who choose to keep their cats indoors may be aware of this. It is also possible that the increased interactions between cat and owner for indoor cats may lead to a closer perceived relationship by the owner, who can get to know their cat very well due to the time spent together. However, this study cannot determine the direction of causality, and it is equally possible that people who perceive a strong emotional connection with their cat choose to keep the cat indoors so that they can be together as often as possible.

The POI and PC subscales were both influenced by the presence of other pets in the home. Having dogs in the home was associated with reduced scores on both subscales. This means that cat owners who also own dogs report having fewer interactions with their cat but are also less likely to perceive that their cat is expensive, either financially or in terms of time costs. This result can likely to be explained by owners comparing their dogs and cats when completing this survey. Dogs have more daily requirements (e.g., regular walks, trips to a park or in the car) than cats, which may reduce the time available for the owner to spend with the cat. Dogs are also more expensive to maintain than cats. For example, recent research has found that dog owners spent an average of AUD 3200 (EUR 2100; USD 2220) on their dog between March 2020 and March 2021, compared to AUD 2100 (EUR 1400; USD 1400) for cats [40]. Therefore, dog ownership probably changes cat owners’ perceptions of how expensive their cat is, as well as how much time they spend with their cat, compared to people who only own cats.

There were a few other demographic factors influencing the PC subscale. Owners with less experience owning previous cats were more likely to perceive that their cat was expensive compared to people with lots of previous experience. Similarly, owners of neutered cats perceived fewer costs than owners of unneutered cats, owners of female cats perceived their pet’s care to be less expensive than owners with male cats, and cats with behavioural problems were more likely to be perceived as expensive than those without such problems. Again, the direction of causality, if there is any, is unknown. Perhaps people who own cats with behavioural problems find the cat expensive because of the behavioural problems, or perhaps lack of adequate cat management, due to high perceived costs of looking after the cat, led to the cat having behaviour problems.

The study has some limitations: first, being questionnaire-based research, there might be a bias on participants that joined the study. Owners that are more mindful of their relationship with their pets, are also more likely to participate. Second, as it often happens in studies that address the human–animal relationship [41,42], our sample was mainly composed of female owners. Future research should aim to obtain a representative sample of the population to better understand the subtleties of cat-owner relationships.

## 5. Conclusions

The Italian version of the C/DORS for cats used in the current study is a valid and reliable tool to investigate the relationship between owners and their cat. To our knowledge this is the first study that administered an Italian version of the C/DORS for cat owners and with the aim to validate it. Owners of cats living exclusively indoors reported higher scores on perceived pet–owner interactions and perceived emotional closeness compared to indoor/outdoor cats. Cat owners who also had dogs reported fewer interactions with their cat, but also fewer costs of cat ownership, than owners without dogs. Finally, behaviour problems, not being neutered, and lack of previous experience with cat ownership were associated with higher perceived costs of having the pet cat.

Future research should focus on administering the C/DORS questionnaire also in other languages to confirm the validity of this scale as a reliable common tool to investigate owner-pet relationship around the world.

## Figures and Tables

**Table 1 animals-13-00069-t001:** Descriptive statistics of the scales used in the study.

Variable	Mean (Min-Max)	Standard Deviation	Skewness	Kurtosis	Shapiro–Wilk Normality Test
**Cat/Dog–Owner Relationship Scale**					
Pet–Owner Interactions	4.86 (1.38–7)	0.8	−0.08	1.15	0.97 (<0.001)
Perceived Emotional Closeness	5.5 (2–7)	0.98	–0.56	–0.08	0.97 (<0.001)
Perceived Costs	6.04 (3–7)	0.78	–0.78	0.11	0.93 (<0.001)
**Lexington Attachment to Pets Scale**					
General Attachment	37.05 (16–44)	4.73	–0.58	0.1	0.96 (<0.001)
Animal Rights	17.79 (9–20)	2.21	–1.12	0.82	0.87 (<0.001)
People substitute	19.22 (7–28)	4.75	–0.04	–0.7	0.98 (<0.001)

**Table 2 animals-13-00069-t002:** Demographic information of the participants.

Demographic Factor	% (N)
**Gender**	
Female	88.1 (1040)
Male	11.4 (135)
Other/Prefer not to say	0.5 (1)
**Level of Education**	
Lower education	2.9 (34)
Up to secondary degree	35 (414)
Tertiary degree and above	62.1 (735)
**Occupation**	
Working with animals	11.9 (139)
Not working with animals /Unemployed	76.6 (897)
Freelancer	0.1 (1)
Retired	5.9 (69)
Student	5.6 (65)
**Geography**	
City (population 20,000+)	57.9 (685)
Town (Population 2000–20,000)	32 (379)
Village (population less than 2000)	4.9 (58)
Isolated (es. Countryside)	5.2 (61)
**Household composition**	
Single	22.1 (260)
Couple	42.3 (498)
3 people	19.8 (233)
4+ people	15.7 (185)
**Number of Previous cats**	
0	19.0 (224)
1	15.9 (188)
2	13.3 (157)
3	10.5 (124)
4+	41.2 (486)
**Previous experience with cats**	
Had the first cat at younger than 10 years old	44.7 (529)
Had the first cat between 10 and 20 years old	20.4 (241)
Had the first cat at older than 20 years old	34.9 (413)

**Table 3 animals-13-00069-t003:** Results of the C/DORS’ EFA and factor loadings. The highest loading for each item is in bold.

Items	Loadings		
	Pet-Owner Interactions (POI)	Perceived Emotional Closeness (PEC)	Perceived Costs (PC)
26 How often do you pet your cat?	**0.85**	−0.08	−0.03
21 How often do you cuddle your cat?	**0.84**	−0.04	0.01
15 How often do you talk to your cat?	**0.66**	0.03	0.06
9 How often do you spend time enjoying watching your cat?	**0.63**	0.01	0.05
30 How often do you hug your cat?	**0.59**	0.22	−0.08
4 How often do you kiss your cat?	**0.46**	0.26	−0.02
23 How often do you have your cat with you while relaxing, e.g., watching TV?	**0.44**	0.22	0.03
7 How often do you play games with your cat?	**0.35**	0.12	0.03
24 My cat is there whenever I need to be comforted.	−0.052	**0.85**	−0.02
18 If everyone else left me, my cat would still be there for me.	−0.07	**0.80**	−0.04
20 My cat helps me get through tough times.	0.03	**0.72**	0.002
22 My cat provides me with constant companionship.	0.08	**0.66**	0.04
32 My cat is constantly attentive to me.	−0.03	**0.65**	0.09
17 I would like to have my cat near me all the time.	0.14	**0.55**	0.06
5 I wish my cat and I never had to be apart.	0.14	**0.50**	0.08
25 How traumatic do you think it will be for you when your cat dies?	0.13	**0.44**	0.002
2 My cat gives me a reason to get up in the morning	0.06	**0.41**	0.07
13 How often do you tell your cat things you don’t tell anyone else?	0.23	**0.33**	−0.02
10 It is annoying that sometimes I have to change my plans because of my cat.	−0.001	−0.01	**0.74**
8 It bothers me that my cat stops me doing things I enjoyed before I owned it.	−0.04	0.02	**0.69**
6 My cat makes too much mess.	0.02	−0.04	**0.56**
11 My cat costs too much money.	0.03	−0.04	**0.50**
3 There are major aspects of owning a cat I don’t like.	−0.01	0.14	**0.43**
1 How hard is it to look after your cat?	0.04	0.03	**0.42**
27 How often do you take your cat to visit people?	-	-	-
29 How often do you take your cat in the car?	-	-	-
12 How often do you buy your cat presents?	-	-	-
14 How often do you feel that looking after your cat is a chore?	-	-	-
16 How often does your cat stop you doing things you want to?	-	-	-
19 How often do you feel that having a cat is more trouble than it’s worth?	-	-	-
28 How often do you give your cat food treats?	-	-	-
31 How often do you groom your cat?	-	-	-
	**POI**	**PEC**	**PC**
**Variance explained**	17%	14%	8%
**Cronbach’s alpha**	0.84	0.85	0.71

**Table 4 animals-13-00069-t004:** Test–retest reliability (n = 30) with median and range scores for the first participation (T0) and the second (T1).

Item	T_0_ Median (Range)	T_1_ Median (Range)	Spearman Rho	*p*
**Lexington Attachment to Pets Scale**				
General Attachment	36 (21–44)	35.5 (20–44)	0.87	<0.001
Animal Rights	18.5 (8–20)	18.0 (7–20)	0.89	<0.001
People substitute	18 (8–26)	18.5 (8–26)	0.79	<0.001
**Cat/Dog-Owner Relationship Scale**				
Pet-Owner Interactions	35.5 (28–51)	35.5 (22–48)	0.78	<0.001
Perceived Emotional Closeness	49.5 (12–17)	48.5 (15.65)	0.87	<0.001
Perceived Costs	10.0 (6–19)	12.0 (6–21)	0.67	<0.001

**Table 5 animals-13-00069-t005:** Spearman’s correlations between LAPS and C/DORS and between C/DORS domains (Spearman rho, with *p* values in brackets).

	Pet–Owner Interactions	Perceived Emotional Closeness	Perceived Costs
General Attachment	0.51 (<0.001)	0.75 (<0.001)	0.28 (<0.01)
Animal Rights	0.36 (<0.001)	0.54 (<0.001)	0.26 (<0.001)
People substitute	0.44 (<0.001)	0.71 (<0.001)	0.21 (<0.001)
Pet–Owner Interactions	-	0.59 (<0.001)	0.15 (<0.001)
Perceived Emotional Closeness	-	-	0.31 (<0.001)
Perceived Costs	-	-	-

Note: A Bonferroni correction has been calculated for 12 tests, with alpha level set at 0.001.

**Table 6 animals-13-00069-t006:** Variable selections for the C/DORS regression model. Only the variables included in the final models are in bold.

Predictors	Pet-Owner Interactions	Perceived Emotional Closeness	Perceived Costs
Owner’s age ^a^	r = −0.46, *p* = 0.14	r = 0.0006, *p* = 0.99	r = 0.08, *p* = 0.01
Owner’s gender ^b^	Χ^2^ (df = 2) = 7.05, *p* = 0.03	Χ^2^ (df = 2) = 14.25, *p* ≤ 0.001	Χ^2^ (df = 2) = 15.26, *p* ≤ 0.001
Level of education ^b^	Χ^2^ (df = 2) = 3.43, *p* = 0.18	Χ^2^ (df = 2) = 3.87, *p* = 0.14	Χ^2^ (df = 2) = 7.47, *p* = 0.02
Occupation ^b^	Χ^2^ (df =4) = 16.60, *p* = 0.002	Χ^2^ (df = 4) = 11.82, *p* = 0.02	Χ^2^ (df = 4) = 4.59, *p* = 0.33
Geography ^b^	Χ^2^ (df =3) = 9.08, *p* = 0.03	Χ^2^ (df = 3) = 5.50, *p* = 0.14	Χ^2^ (df = 3) = 9.80, *p* = 0.02
Household composition ^b^	Χ^2^ (df = 3) = 21.26, *p* ≤ 0.001	**Χ^2^ (df = 3) = 21.26, *p* ≤ 0.001**	Χ^2^ (df = 3) = 0.20, *p* = 0.98
Number of previous cats ^b^	Χ^2^ (df = 4) = 1.46, *p* = 0.83	Χ^2^ (df = 4) = 7.23, *p* = 0.12	**Χ^2^ (df = 4) = 25.48, *p* ≤ 0.001**
Previous experience with cats ^b^	Χ^2^ (df = 2) = 1.08, *p* = 0.58	Χ^2^ (df = 2) = 5.68, *p* = 0.06	Χ^2^ (df = 2) = 12.76, *p* = 0.002
Cat’s age ^a^	r = −0.05, *p* = 0.14	r = 0.04, *p* = 0.24	r = 0.13, *p* ≤ 0.001
Cat’s age at adoption ^a^	r = −0.001, *p* = 0.97	r = −0.01, *p* = 0.87	r = −0.003, *p* = 0.94
Cat’s sex ^c^	W = 117,477, *p* = 0.26	W = 115,752, *p* = 0.13	W = 129.396, *p* = 0.01
Neutering status ^c^	W = 15,905, *p* = 0.62	W = 16,108, *p* = 0.71	W = 12.915, *p* = 0.03
Age when neutered ^b^	Χ^2^ (df = 4) = 4.82, *p* = 0.31	Χ^2^ (df = 4) = 3.19, *p* = 0.53	Χ^2^ (df = 4) = 8.25, *p* = 0.08
Weight ^a^	r = 0.05, *p* = 0.10	r = 0.09, *p* = 0.01	r = 0.03, *p* = 0.28
Breeds ^b^	Χ^2^ (df = 12) = 16.50, *p* = 0.17	Χ^2^ (df = 12) = 10.41, *p* = 0.58	Χ^2^ (df =12) = 16.69, *p* = 0.16
Origin of the cat ^b^	Χ^2^ (df = 6) = 6.59, *p* = 0.36	Χ^2^ (df =6) = 10.84, *p* = 0.09	Χ^2^ (df = 6) = 5.23, *p* = 0.52
Activities with the cat ^b^	Χ^2^ (df = 3) = 20.50, *p* ≤ 0.001	Χ^2^ (df = 3) = 16.11, *p* = 0.001	Χ^2^ (df = 3) = 4.14, *p* = 0.25
Health issues ^c^	W = 85,012, *p* = 0.56	W = 80,514, *p* = 0.53	W = 84,825, *p* = 0.11
Behaviour issues ^c^	W = 47,678, *p* = 0.053	W = 46,961, *p* = 0.10	**W = 53.036, *p* ≤ 0.001**
Living space ^b^	**Χ^2^ (df = 5) = 26.72, *p* ≤ 0.001**	**Χ^2^ (df = 5) = 21.25, *p* = 0.001**	Χ^2^ (df = 5) = 10.42, *p* = 0.06
Other pets ^b^	**Χ^2^ (df = 3) = 35.91, *p* ≤ 0.001**	Χ^2^ (df = 3) = 21.27, *p* ≤ 0.001	**Χ^2^ (df = 3) = 15.46, *p* = 0.001**
Perceived cat’s attachment ^b^	**Χ^2^ (df = 6) = 135.66, *p* < 0.001**	**Χ^2^ (df = 6) = 135.65, *p* < 0.001**	**Χ^2^ (df = 6) = 47.31, *p* < 0.001**

Note: bold = *p* < 0.001, ^a^ = Spearman correlation, ^b^ = Kruskall–Wallis test, ^c^ = Mann–Whitney U test.

**Table 7 animals-13-00069-t007:** Ordinal regressions showing a summary of the most relevant response variables C/DORS Pet–Owner Interactions, Perceived Emotional Closeness and Perceived Costs.

Dependent Variable	Parameter	B	Std. Error	Sig	Exp (B)
Pet–Owner Interactions	[Living space = outdoors and indoors Vs. indoors only]	−0.47	0.14	0.01	−3.37
	[Other pets = both dogs and cats Vs. alone]	−0.48	0.17	0.02	−2.93
	[Other pets = both dogs and cats Vs. only cats]	−0.50	0.16	0.01	−3.12
	[Other pets = alone Vs. only dogs]	0.71	0.20	0.002	3.63
	[Other pets = only dogs Vs. only cats]	−0.72	0.19	0.001	−3.78
Perceived Emotional Closeness	[Living space = indoors only Vs. outdoors and indoors]	−0.33	0.08	<0.001	−4.20
Perceived Cost	[N. of previous cats = 0 Vs. 1]	−0.38	0.09	0.006	−4.00
	[N. of previous cats = 1 Vs. 4 or more]	−0.35	0.09	0.003	−3.58
	[Behaviour problems = no Vs. yes]	0.55	0.11	<0.001	4.85
	[Other pets = alone Vs. only dogs]	−0.44	0.12	0.001	−3.79
	[Other pets = only dogs Vs. only cats]	0.42	0.11	0.001	3.68

Significance: *p* < 0.05 (only parameters with *p* < 0.05 are reported). B: regression coefficient. SE: standard error of the mean.

## Data Availability

The data presented in this study are available on request from the corresponding author.

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
