# Peer review of "The Cat–Owner Relationship: Validation of the Italian C/DORS for Cat Owners and Correlation with the LAPS"

_animals, 2022, doi:10.3390/ani13010069_

Round 1
Reviewer 1 Report
The paper is adequate and well organized, however, it is important to include a table with the removed items, including as much information as possible (Alphas, betas, skewness, kurtosis, mean, median and SD) and also, to mentione on which factor the items are loaded. It is necessary to show the factor analysis (EFA), before the items were removed. Then, explain a bit more why those items did not load on the corresponding subscale. Also, discuss the theoretical importance of the removed items.
On the other hand, it is convenient that the first model and the final one (table 3) are compared with more researches. In addition, it is convenient to show the complete scale in its Italian version as an annex.
Author Response
The paper is adequate and well organized, however, it is important to include a table with the removed items, including as much information as possible (Alphas, betas, skewness, kurtosis, mean, median and SD) and also, to mentione on which factor the items are loaded. It is necessary to show the factor analysis (EFA), before the items were removed. Then, explain a bit more why those items did not load on the corresponding subscale. On the other hand, it is convenient that the first model and the final one (table 3) are compared with more researches.
Thank you for the suggestion we included the results in the supplementary material Table S2 and Table S3
Also, discuss the theoretical importance of the removed items.
Thank you for your suggestion we already discussed theoretical implications of some removed items (i.e. lines 407-415; 422-433) but we decided to further discuss them (lines 438-449)
In addition, it is convenient to show the complete scale in its Italian version as an annex.
Thank you for the suggestion, we already provide the complete scale in English and in Italian in the supplementary materials, but we moved everything in the appendix (Table A1) as suggested.
Reviewer 2 Report
Borrelli et al.’s manuscript (animals-2089692) is well-written and addresses whether an Italian version of the C/DORS is valid for measuring the cat-owner relationship. The C/DORS is a valuable research tool and having it available in Italian is important for researchers in Italy and other countries where Italian is an official or common language. That said, because the population of Italian speaking people is relatively small, the primary purpose of the article will have a limited impact. Fortunately, Borrelli et al. include additional results (e.g. comparing pet-owner interactions and perceived emotional closeness for cats living indoors vs. indoor/outdoor cats) that will have a broader impact.
I have two minor issues for the authors to address:
Table 6: That is a lot of statistical tests! How are you protecting yourself from making Type I errors with that many tests?
Lines 418-419: Can you provide a reference to support that cats are neophobic? A popular, and perhaps incorrect, conception of cats is that they are curious which seems to imply that they like to explore new and novel things. Cats may be cautious about approaching new things, but that is different from being fearful of new things.
Additionally, I noticed a few minor typographical or wording issues:
Line 29: “We a higher score…” is awkward.
Line 33: “Nowadays most people globally own a pet.” is awkward. Consider “Globally, most people now own a pet.”
Line 187: “iin addition” should be “in addition”
Line 198: It isn’t clear whether the exclusion criteria are either low standard deviation or high skewness and kurtosis or both low standard deviation and high skewness and kurtosis.
Line 205: Do you mean “oblique” for “obliquus”?
Line 365: “that take care of…” should be “that taking care of…”
Line 425: Consider replacing “subjects” with “cats”.
Author Response
Borrelli et al.’s manuscript (animals-2089692) is well-written and addresses whether an Italian version of the C/DORS is valid for measuring the cat-owner relationship. The C/DORS is a valuable research tool and having it available in Italian is important for researchers in Italy and other countries where Italian is an official or common language. That said, because the population of Italian speaking people is relatively small, the primary purpose of the article will have a limited impact. Fortunately, Borrelli et al. include additional results (e.g. comparing pet-owner interactions and perceived emotional closeness for cats living indoors vs. indoor/outdoor cats) that will have a broader impact.
I have two minor issues for the authors to address:
Table 6: That is a lot of statistical tests! How are you protecting yourself from making Type I errors with that many tests?
Thank you for noticing, we used Bonferroni correction which yielded an alpha level of 0.002, so we used the standard threshold of 0.001. As we inserted the sentence about Bonferroni, we noticed that two variables on the Costs regression were above threshold so we removed them and corrected the text. (Lines 239-240)
Lines 418-419: Can you provide a reference to support that cats are neophobic? A popular, and perhaps incorrect, conception of cats is that they are curious which seems to imply that they like to explore new and novel things. Cats may be cautious about approaching new things, but that is different from being fearful of new things.
Thank you for the comments we changed the sentence at lines 418-419 to avoid misunderstanding (now at lines 422-423).
Additionally, I noticed a few minor typographical or wording issues:
Line 29: “We a higher score…” is awkward.
We corrected the sentence with: “We found a higher …” (line 29)
Line 33: “Nowadays most people globally own a pet.” is awkward. Consider “Globally, most people now own a pet.”
We thank the reviewer for the suggestion. We corrected the sentence as recommended (line 33)
Line 187: “iin addition” should be “in addition”
Thank you for noticing the misspelling, we corrected it (line 187)
Line 198: It isn’t clear whether the exclusion criteria are either low standard deviation or high skewness and kurtosis or both low standard deviation and high skewness and kurtosis.
Thank you for the suggestion, we added the information. (Line 199-204)
Line 205: Do you mean “oblique” for “obliquus”?
Thank you for noticing the misspelling, we corrected it (Line207)
Line 365: “that take care of…” should be “that taking care of…”
Thank you for the comment, we corrected the sentence as suggested (Line 369)
Line 425: Consider replacing “subjects” with “cats”.
Thank you for the comment, we changed the term as suggested (Line 430)